# Chitosan Fibers Loaded with Limonite as a Catalyst for the Decolorization of Methylene Blue via a Persulfate-Based Advanced Oxidation Process

**DOI:** 10.3390/polym14235165

**Published:** 2022-11-27

**Authors:** Siew Xian Chin, Kam Sheng Lau, Sarani Zakaria, Chin Hua Chia, Chatchawal Wongchoosuk

**Affiliations:** 1Department of Physics, Faculty of Science, Kasetsart University, Chatuchak, Bangkok 10900, Thailand; 2ASASIpintar Program, Pusat GENIUS@Pintar Negara, Universiti Kebangsaan Malaysia, Bangi 43600, Selangor, Malaysia; 3Materials Science Program, Department of Applied Physics, Faculty of Science and Technology, Universiti Kebangsaan Malaysia, Bangi 43600, Selangor, Malaysia

**Keywords:** advanced oxidation process, decomposition, iron mineral, wastewater treatment

## Abstract

Wastewater generated from industries seriously impacts the environment. Conventional biological and physiochemical treatment methods for wastewater containing organic molecules have some limitations. Therefore, identifying other alternative methods or processes that are more suitable to degrade organic molecules and lower chemical oxygen demand (COD) in wastewater is necessary. Heterogeneous Fenton processes and persulfate (PS) oxidation are advanced oxidation processes (AOPs) that degrade organic pollutants via reactive radical species. Therefore, in this study, limonite powder was incorporated into porous regenerated chitosan fibers and further used as a heterogeneous catalyst to decompose methylene blue (MB) via sulfate radical-based AOPs. Limonite was used as a heterogeneous catalyst in this process to generate the persulfate radicals (SO_4_^−^·) that initiate the decolorization process. Limonite–chitosan fibers were produced to effectively recover the limonite powder so that the catalyst can be reused repeatedly. The formation of limonite–chitosan fibers viewed under a field emission scanning electron microscope (FESEM) showed that the limonite powder was well distributed in both the surface and cross-section area. The effectiveness of limonite–chitosan fibers as a catalyst under PS activation achieved an MB decolorization of 78% after 14 min. The stability and reusability of chitosan–limonite fibers were evaluated and measured in cycles 1 to 10 under optimal conditions. After 10 cycles of repeated use, the limonite–chitosan fiber maintained its performance up to 86%, revealing that limonite-containing chitosan fibers are a promising reusable catalyst material.

## 1. Introduction

Rapid industrialization, which utilizes organic dyes in the agriculture, food technology, leather, and textile industries, has caused major environmental issues [1]. Several technologies have been employed to tackle the wastewater issues, such as physical adsorption methods, chemically advanced oxidation processes (AOPs), biodegradation approaches, and traditional flocculation and coagulation methods [2,3,4]. Among all of these, AOPs have gained popularity due to their ability to degrade non-biodegradable organic molecules, which can lower the chemical oxygen demand (COD) and biological oxygen demand (BOD) of the treated wastewater [5].

Recently, polysaccharide-based polymers have been used to develop materials for wastewater treatment [6]. Particularly, chitosan and its derivatives have attracted wide attention due to many desired properties, including biodegradability, renewability, being an abundant resource, being non-toxic, and having high functionalities [7]. Previously, various forms of chitosan products, such as aerogels [8], beads [9], fibers [10], membranes [11], and hydrogel [12], were successfully produced by incorporating other materials to enhance their performance in wastewater treatment.

AOPs utilize powerful free radicals such as hydroxyl radicals (OH·) and sulfate radicals (SO_4_^−^·), generated via homogenous or heterogenous catalysis, photocatalysis, electrocatalysis, etc, which decompose organic molecules in water (mineralization to CO_2_, H_2_O, and inorganic ions) [13,14,15]. The Fenton process is a popular OH·-based AOP that uses iron species (Fe ions/oxides) as the catalyst to generate OH· radicals from hydrogen peroxide (H_2_O_2_) [16]. However, OH·-based AOPs possess limitations, including the requirement of acidic conditions (pH 2–4) and H_2_O_2_ instability [17]. Preferences have been shifted towards SO_4_^−^· radicals instead of OH· radicals due to their higher redox potential and the stability of the SO_4_^−^· radicals, which are beneficial for the degradation of organic compounds [18,19].

Limonite is a type of natural iron ore that contains hydrated iron oxides, such as, goethite, hematite, and magnetite [20]. Limonite has also been used in various applications, such as the catalytic cracking process [21], wastewater treatments [22,23], radiation shielding materials [24], and corrosion resistance additives [25]. Previous studies also show that limonite is an efficient catalyst for Fenton/H_2_O_2_ reactions to degrade organic molecules [26,27]. However, its potential in AOPs reactions is yet to be discovered.

In this study, we prepared regenerated chitosan fibers containing various amounts of limonite and further used them as a recyclable catalyst to decolorize methylene blue in an aqueous solution via SO_4_^−^·-based AOPs. Immobilization of limonite powder into the chitosan fibers offers several advantages, such as the easy recovery and separation of the catalyst [28]. Methylene blue was chosen as an organic model compound to investigate the decolorizing efficiency of the chitosan–limonite fibers via SO_4_^−^·-based AOPs. The use of chitosan in this study is due to its abundance in resources and its ease of incorporation into different products using acetic acid via dissolution and regeneration processes.

## 2. Materials and Methods

### 2.1. Materials

Chitosan (medium molecular weight, 75–85% deacetylated), sodium persulfate (PS, Na_2_S_2_O8, >98%), and methylene blue (C_16_H_18_ClN_3_S, ≥95%) were obtained from Sigma-Aldrich. Glacial acetic acid (CH_3_COOH, 100%, EMSURE^®^ ACS), hydrochloric acid (HCl, 37%, EMSURE^®^ ACS), and sodium hydroxide (NaOH, 99–100%, EMSURE^®^ ACS) were obtained from Merck Millipore.

### 2.2. Preparation of Chitosan Fibers Loaded with Limonite

A 2 wt% of chitosan dissolved in 2.5 wt% acetic acid was prepared by heating the mixture solution using a hotplate at 90 °C for 1 h. The limonite powder was added to the chitosan solution with different weight percentages (1%, 10%, and 20%) and then stirred for 1 h using a magnetic stirrer to homogenize the mixture. The chitosan–limonite mixture was then loaded into a syringe attached with a 21G needle (inner diameter 0.51 mm) and then the solution was injected manually into a 50 mL NaOH (0.1 M) solution to yield regenerated chitosan–limonite fibers continuously. Next, the resulting fibers were washed with deionized water three times to remove excess chemicals.

The chitosan fibers loaded with limonite were characterized using Fourier transform infrared (FTIR) spectrometer (Bruker, Alpha, Ettlingen, Germany) measured from 4000–600 cm^−1^ with an interval of 4 cm^−1^. The morphology of the chitosan fibers was investigated by a field emission scanning electron microscope (FESEM, Schottky SU5000, Tokyo, Japan). X-ray diffraction (XRD) was conducted using an X-ray diffractometer (Bruker, D8 Advance, Baden, Switzerland) with a monochromatic Cu Kα radiation source (λ = 0.1539 nm), a scan rate of 2°/min, and a 2θ scan range from 10° to 80°.

### 2.3. Adsorption and Decolorization of MB Using Chitosan Fibers Loaded with Limonite

Before the SO_4_^−^·-based AOPs experiment to decolorize MB in an aqueous solution, the adsorption behaviour of the chitosan fibers was investigated in terms of the adsorption isotherm using Langmuir and Freundlich models. Briefly, 3 g of chitosan fibers was added to 50 mL of various MB solutions (5, 10, 25, 50, and 100 mg/L) and incubated at 25 °C for 24 h. Then, the MB solution was filtered using a nylon membrane (0.22 µm) and measured using a UV–vis spectrometer (Jenway, 7315, Chennai, India) to determine the concentration of MB in the solution at a wavelength of 660 nm. The adsorption capacity (q) of the chitosan-limonite fibers was calculated using Equation (1).
(1)$$q=\frac{\left(C_{i}-C_{e}\right)V}{W}$$
where C_e_ is the equilibrium concentration (mg/L), C_i_ is the initial concentration (mg/L), V is the volume of the MB solution taken (mL), and W is the weight of the chitosan fibers (g). The Langmuir and Freundlich isotherms were also used to explain the surface characteristics of the adsorbents, adsorption mechanism, and adsorbent affinity of the chitosan fibers. The linearized Langmuir and Freundlich adsorption isotherms were fitted using Equations (2) and (3).
(2)$$\frac{C_{e}}{q_{e}}=\frac{C_{e}}{q_{m}}+\frac{1}{{\mathrm{bq}}_{m}}$$
(3)$$\ln q_{e}=\ln K_{F}-\frac{1}{n}\ln C_{e}$$
where C_e_ is the equilibrium concentration (mg/L), q_e_ is the adsorption capacity at the equilibrium time (mg/g), q_m_ is the maximum adsorption capacity (mg/g), and b is the Langmuir constant (L/mg) in Equation (2), while K_F_ is the Freundlich’s constant (L/g), and n is the heterogeneity factor or adsorption power in Equation (3).

To perform the SO_4_^−^·-based AOPs experiments, 3 g of chitosan fibers were added to 49 mL of MB solution, and then, the concentration of MB was measured until it achieved the adsorption equilibrium. Next, 1 mL of PS was added to the solution to initiate the SO_4_^−^·-based AOPs process. The experiments were conducted at room temperature. The concentration of MB throughout the experiment was recorded online using an optical cable probe (FC-UVIR200-2) immersed in the solution without requiring withdrawing aliquots from the solution. The probe was connected to a light source (AvaLight-DH-S, Avantes, Apeldoorn, Netherlands) and a UV-vis spectrometer (AvaSpec-ULS2048L-USB2-UA-RS, Avantes) to determine the concentration of MB in the solution at a wavelength of 660 nm. The effect of the limonite content, PS concentration (0, 2, 4, and 8 mM), pH (2–9), and recyclability of the chitosan–limonite fibers were investigated in this study. The percentage of MB decolorization was calculated using Equation (4);
(4)$$\mathrm{Decolorization}\;\left(\%\right)=\frac{C_{0}-C_{t}}{C_{0}}\times 100\%$$
where C_t_ is the MB concentration at a given time (mg/L) and C_0_ is the initial MB concentration (mg/L). The rate of MB decolorization was investigated using the first- and second-order reaction kinetic models as shown in Equations (5) and (6), respectively [27].
(5)$$\ln\frac{C_{t}-C_{e}}{C_{0}-C_{e}}=k_{1}t$$
(6)$$\frac{1}{C_{t}-C_{e}}-\frac{1}{C_{0}-C_{e}}=k_{2}t$$
where k_1_ and k_2_ are the rate constants for the first- and second-order reaction kinetic models, respectively.

The reusability of the chitosan-limonite fibers was tested by performing ten consecutive cycles of the SO_4_^−^·-based AOPs experiments using the same experimental set up. The fibers were rinsed three times by deionized water before being used for next cycle.

### 2.4. Computational Analysis

The density functional theory calculation was conducted in Materials Studio using the DMol3 module (Biovia Discovery Studio 2021 (License Keys: LKO 1986646) associated with Galaxy FCT Sdn Bhd under Dassault Systèmes’ 3DEXPERIENCE Lab accelerator program). The generalized gradient approximation (GGA) was used in the scheme of Perdew–Burke–Ernzerhof (PBE) to describe the exchange-correlation function. The chemical structures of goethite, hematite, and PS underwent geometry optimization tasks with a convergence tolerance of energy 1 × 10^−5^ Hartree, a force of 0.002 Hartree/Å, and a displacement of 0.005 Å. The basis set was set as the DNP with a basis file of 3.5. The global orbital cutoff was set at 4.6 Å. The adsorption energy (E_ads_) was calculated using Equation (7):(7)$$E_{\mathrm{ads}}=E_{\mathrm{sys}}-\left(E_{\mathrm{adsorbent}}+E_{\mathrm{adsorbate}}\right)$$
where E_sys_, E_adsorbent_, and E_adsorbate_ are the total energies of the iron oxide slab with the PS molecule, iron oxide slab, and PS molecule, respectively.

## 3. Results and Discussion

### 3.1. Characterization of the Chitosan-Limonite Fibers

The morphology of the chitosan–limonite fibers was examined from a cross-section and surface perspective, as shown in Figure 1. The limonite powders are well distributed on the surface of the chitosan fiber and the internal section of the chitosan fiber (Figure 1a). As can be seen at higher magnifications, the cross-section of the chitosan fiber is more porous compared to its surface. The pore size distribution of the cross-section of the chitosan fiber was estimated at around 0.23–0.51 μm. The EDX results show that the chitosan-limonite fibers contained 43.2%, 4.1%, 30.7%, 0.1%, 0.3%, and 21.5% of the elements C, N, O, Al, Si, and Fe, respectively, indicating the successful incorporation of limonite into the chitosan fibers.

The FTIR spectra of the chitosan fiber containing different contents of limonite are shown in Figure 2a. Typically, the hygroscopic nature of chitosan that adsorbs environmental moisture gives rise to the broad peak at around 3500 to 3000 cm^−1^ corresponding to O-H. The other peaks observed at 2870 cm^−1^ (C-H stretching), 1640 and 1550 cm^−1^ (C=O and N-H stretching of amide group), 1370 cm^−1^ (primary alcohol), and 1023 cm^−1^ (C-O-C stretching) correspond to the functional groups of the glucosamine structure in the backbone of the chitosan chain. Figure 2b illustrates the XRD patterns of the chitosan fibers with 10% limonite and the raw limonite powder. There are three major iron compounds present in limonite, including goethite, hematite, and magnetite. The corresponding diffraction peaks for goethite were at 2θ = 21.23°, 47.0°, 58.9°, and 61.3°; hematite at 2θ = 33.2°, 36.7°, and 41.3°; whereas magnetite only showed one diffraction peak at around 2θ = 53.2°. Moreover, the 10% chitosan fiber with 10% limonite showed an amorphous region (blue region) of chitosan polymer chains at around 16.7–23.7°, and the diffraction peaks for the limonite powder also showed the presence of goethite, hematite, and magnetite in the fiber, indicating that the dissolution and regeneration of chitosan processes did not change the crystal phase of the limonite powder.

### 3.2. Adsorption and SO_4_^−^·-Based AOPs Decolorization of MB Using Chitosan-Limonite Fibers

It is anticipated that the removal of MB by the chitosan–limonite fibers involves simultaneous adsorption of MB onto the fibers and degradation of MB by the SO_4_^−^· radicals. Therefore, the adsorption behavior of MB on the fibers was investigated at different concentrations of MB without the addition of PS. The adsorption results and data fitted with Langmuir and Freundlich isotherms are shown in Figure 3. The adsorption isotherm parameters, as tabulated in Table 1, show that the chitosan–limonite fibers are well fitted with Langmuir isotherm, suggesting monolayer adsorption of MB molecules on the chitosan–limonite fibers [29]. The 20% limonite sample showed the highest maximum adsorption capacity (5.9 mg/g) among all the samples due to the negative charge of limonite, contributing to the adsorption of cationic MB [26]. Chitosan fibers with limonite content greater than 20% were not successfully produced due to the difficulty of the injection using the syringe needle.

The decolorization kinetics of MB via SO_4_^−^·-based AOPs was conducted using the chitosan–limonite fibers to study the effect of limonite content and PS concentration (Figure 4). The decolorization percentage of MB for the chitosan fibers with 1, 10, and 20% limonite content were of 73, 79, and 75%, respectively. Additional decolorization experiments were also conducted using 100% limonite powder (without chitosan) with the same solid weight of the chitosan–limonite fibers, i.e., 0.03 g, 0.3 g, and 0.6 g limonite. The results are shown in Appendix A. It shows a similar trend to when using the chitosan–limonite fibers, but a slightly lower decolorization percentage. This can be attributed to the agglomeration of the limonite particles as they were suspended into the MB solution during the AOPs experiment. This observation has further proven that the chitosan helps in avoiding agglomeration of limonite, increasing the exposed surface area of the limonite particles. Besides, immobilization of limonite powder into the chitosan fibers offers other advantages, including avoiding leaching of the limonite powder and being easy to recollect the fibers especially for the utilization in larger scale, such as packing in a fixed bed column.

The first- and second-order kinetics models were employed to investigate the decolorization rate of MB, and the results are tabulated in Table 2. The chitosan fibers with 10% limonite showed the highest rate constant among all the chitosan fibers, with 0.402 min^−1^ and 2.974 min^−1^ for the first- and second-order reaction kinetic models, respectively. This was attributed to the optimum molar ratio, which was approximately 10:3, for the PS:Fe_2_O_3_ content present in the experiment medium. Previous studies show that the optimum molar ratio of PS:iron oxide is 1:2 for the degradation of methyl orange and sulfamonomethoxine, as reported by Zhu et al. [29] and Yan et al. [30], respectively. However, Shang et al. reported a higher molar ratio (10:1) of PS:Fe^2+^ for the degradation of diatrizoate [31], while Al-Shamsi and Thomson reported that PS activated by nanoscale zerovalent iron had an optimum molar ratio of 1:1 [32]. The difference in the ratio of PS and the iron-based catalysts used in different studies could be due to factors, such as (1) formation of the iron catalyst compound, (2) size or exposure surface area of the iron catalyst, (3) pH of the medium, (4) temperature, etc. In fact, each variable can interact with one another. Therefore, optimization of the catalytic efficiency should be conducted for each catalyst.

The chitosan–limonite fibers with 10% limonite were used to study the effect of PS concentration on the MB decolorization, and the results are shown in Figure 4d. The reaction without PS also showed a slight decolorization of MB (~10%), which could be due to the adsorption of MB onto the chitosan–limonite fibers as a result of the opposite charges of the limonite and MB. The decolorization of MB using 4 mM PS yielded the highest percentage and rate of decolorization compared to the other concentrations of PS. The decolorization kinetics also fitted with the first-order kinetic model, which showed a better regression value, suggesting that PS concentration is the main factor affecting the decolorization efficiency. Excessive PS could reduce the generation of SO_4_^−^· radicals by the reactant elimination occurring between the SO_4_^−^· radicals and S_2_O_8_^2−^ ions to form SO_4_^2−^ ions, leading to a low decolorization efficiency [33].

Furthermore, the effect of MB concentration on the decolorization via SO_4_^−^·-based AOPs showed that the higher the MB concentration, the greater the percentage of decolorization (Figure 5a). The first- and second-order reaction kinetics models (Figure 5b,c) were also employed to investigate the effect of MB concentration on the decolorization rate of MB. The experiments are better fitted with a first order reaction kinetic model and the reaction rate increased with increasing MB concentration (as shown in Table 2). Therefore, the experimental data were used to determine the dependence of reaction rate on concentrations of MB and PS, and the calculated reaction orders are shown as follows:Rate = k[MB]^0.02^[PS]^0.04^(8)

The reaction order of [MB] shows a positive value (0.02) which indicates that the reaction is MB concentration-dependent.

Moreover, the effect of pH on the MB solution showed that the MB decolorization by SO_4_^−^·-based AOPs was favored at pH 5.78, neutral (pH 7), and alkaline (pH 9.3), with decolorization percentages of 79, 72%, and 69%, respectively. The MB decolorization dropped drastically at a low pH medium (pH 2.7), with 21% of MB decolorization. This could be due to the highly acidic medium deactivating the SO_4_^−^· radicals that form the acid intermediates [34]. The stability and reusability of the chitosan–limonite fibers were evaluated for 10 consecutive cycles for the decolorization of MB at pH 5.78 for 15 min (Figure 5c). The decolorization efficiency was maintained at 86% for up to 10 consecutive cycles relative to the first decolorization percentage, suggesting that the limonite–chitosan fibers are a promising material to be reused as a heterogeneous catalyst for SO_4_^−^·-based AOPs.

### 3.3. Computational Adsorption Energy Calculation of the Adsorption of PS onto Limonite

Based on the XRD results, the two major iron compounds present in the limonite powder are goethite and hematite. Therefore, to further understand which compound plays a greater role in catalyzing the generation of SO_4_^−^· radicals from PS, the crystal structure of goethite and hematite was constructed to calculate the adsorption energy by the PS molecule, as shown in Figure 6. The distance between the O atom in the PS molecule and the Fe atoms in the iron oxide slab was set at 2 Å. The calculated adsorption energy of the PS molecule onto the (020) goethite slab and (104) hematite slab was −1.23 and −1.71 eV, respectively. The higher adsorption energy of the PS onto the (104) hematite slab indicated that the adsorption of the PS molecule was exothermically more favored than the (020) goethite slab [35].

## 4. Conclusions

We successfully synthesized chitosan–limonite fibers loaded with limonite powder and further used the fibers as heterogeneous catalysts for the SO_4_^−^·-based AOPs to decolorize methylene blue. The FESEM and EDX analyses confirmed the formation of porous chitosan fibers with a good distribution of limonite particles throughout the fibers. The addition of limonite into the chitosan fibers showed a higher adsorption capacity toward methylene blue. However, the chitosan fibers with 10% limonite exhibited the highest decolorization percentage (79%) at the optimum of pH 5.8 and 4 mM persulfate. The chitosan–limonite fibers also showed good recyclability up to 10 cycles with a decolorization efficiency greater than 86%. Persulfate was more favorably adsorbed onto hematite than goethite, suggesting that hematite is more dominant in catalyzing the generation of SO_4_^−^· radicals from persulfate. The produced chitosan–limonite fibers could be also potentially used for other oxidation reactions which require iron or iron hydroxide compounds as a heterogeneous catalyst, such as Fenton oxidation.

## Figures and Tables

**Figure 1 polymers-14-05165-f001:**
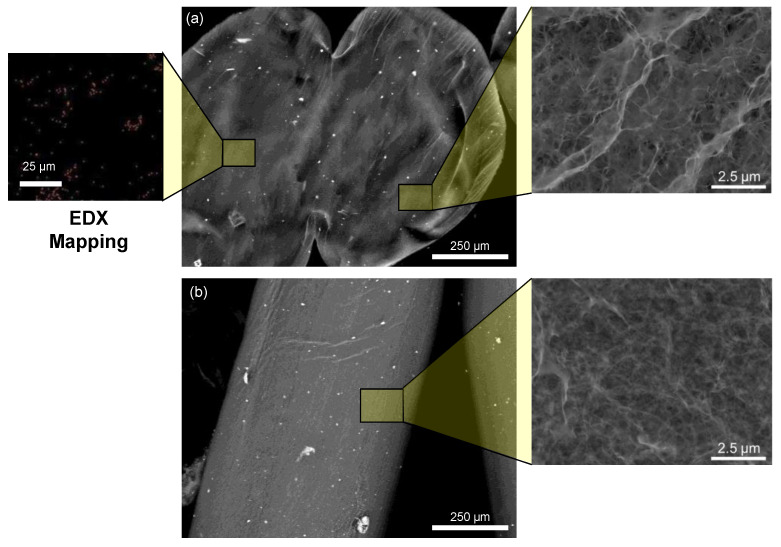
FESEM images of the chitosan–limonite fibers containing 10% limonite: (**a**) cross-section (inset shows the EDX elemental mapping of the fibers representing the distribution of Fe element) and (**b**) surface of the fibers viewed with an enlarged view inserted on the right.

**Figure 2 polymers-14-05165-f002:**
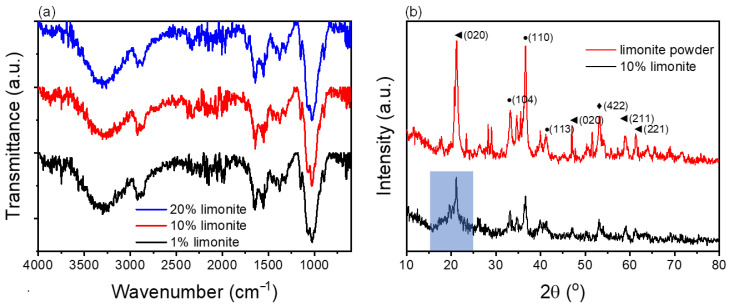
(**a**) FTIR spectra for chitosan fibers containing various amounts of limonite powder. (**b**) XRD diffractogram for the limonite powder and 10% limonite–chitosan fiber with label crystal phase of (◄): goethite, (●) hematite, and (♦) magnetite.

**Figure 3 polymers-14-05165-f003:**
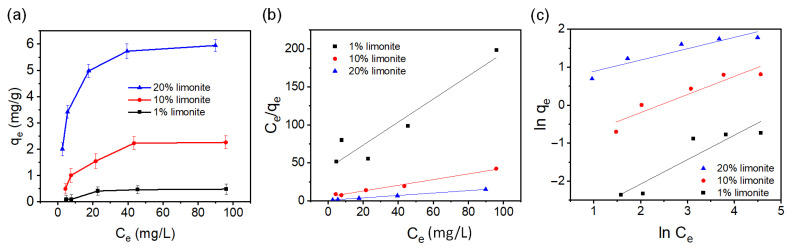
(**a**) Plot of q_e_ vs. C_e_. Adsorption data fitted with (**b**) linearized Langmuir isotherm, and (**c**) linearized Freundlich isotherm.

**Figure 4 polymers-14-05165-f004:**
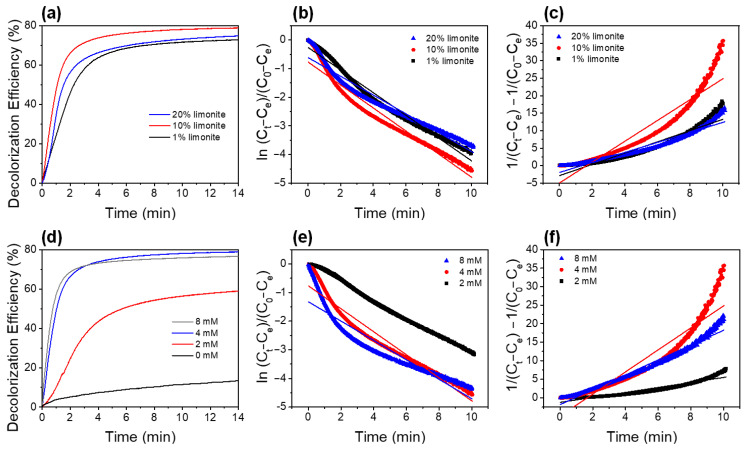
(**a**) Decolorization kinetics of MB with 4 mM PS using chitosan–limonite fibers (1%, 10%, and 20% limonite). Decolorization kinetics data fitted with (**b**) first- and (**c**) second-order reaction kinetic models. (**d**) Decolorization kinetics of MB using chitosan–limonite fibers (10% limonite) and different PS concentrations (2, 4, and 8 mM) on the MB decolorization with 10% limonite content. Decolorization kinetics data fitted with (**e**) first- and (**f**) second-order reaction kinetic models.

**Figure 5 polymers-14-05165-f005:**
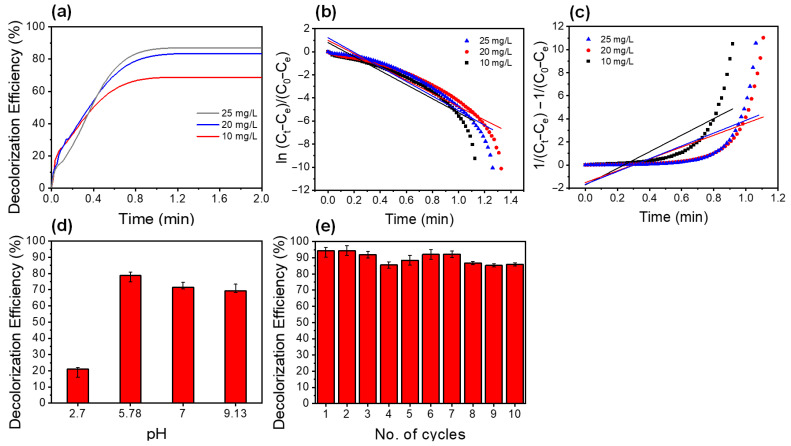
(**a**) Decolorization kinetics of MB at different concentrations of MB using 4 mM PS and chitosan–limonite fibers (10% limonite). Decolorization kinetics data fitted with (**b**) first- and (**c**) second-order reaction kinetic models. (**d**) Effect of pH on the decolorization efficiency (4 mM PS and chitosan–limonite fibers (10% limonite)). (**e**) The recyclability of the chitosan fibers with 10% limonite on the MB decolorization with 4 mM PS and 5 mg/L MB (plotted relative to the first cycle of the decolorization experiment).

**Figure 6 polymers-14-05165-f006:**
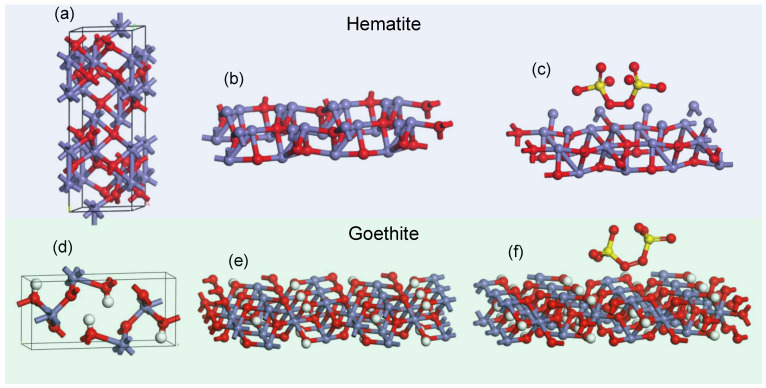
Conventional crystal structure of (**a**) hematite, (**b**) the crystal slab surface of the (104) plane, and (**c**) with PS molecule on the slab surface. Conventional crystal structure of (**d**) goethite, (**e**) the crystal slab surface of the (020) plane, and (**f**) with PS molecule on the slab surface. Atom labels: Fe (blue), O (red), H (white), and S (yellow).

**Table 1 polymers-14-05165-t001:** Langmuir and Freundlich adsorption isotherm parameters for the adsorption of MB.

Sample	Langmuir Isotherm			Freundlich Isotherm		
	q_m_ (mg/g)	b (L/mg)	R^2^	K_F_ ((mg/g) (L/mg)^1/n^)	n	R^2^
1% limonite	0.659	0.0353	0.90	0.0347	1.556	0.86
10% limonite	2.660	0.0691	0.99	0.317	2.101	0.89
20% limonite	6.291	0.2079	0.99	1.807	3.347	0.88

**Table 2 polymers-14-05165-t002:** First- and second-order reaction kinetic model parameters for the decolorization of MB via SO_4_^−^·-based AOPs.

Effect of Limonite Content					Effect of PS Concentration					Effect of MB Concentration				
Limonite Content (wt%)	First-Order Reaction Kinetic Model		Second-Order Reaction Kinetic Model		PS Concentration (mM)	First-Order Reaction Kinetic Model		Second-Order Reaction Kinetic Model		MB Concentration (mg/L)	First-Order Reaction Kinetic Model		Second-Order Reaction Kinetic Model	
	k_1_ (min^−1^)	R^2^	k_2_ (L mg^−1^ min^−1^)	R^2^		k_1_ (min^−1^)	R^2^	k_2_ (L mg^−1^ min^−1^)	R^2^		k_1_ (min^−1^)	R^2^	k_2_ (L mg^−1^ min^−1^)	R^2^
1	0.396	0.98	1.608	0.91	2	0.322	0.99	0.673	0.89	10	7.268	0.94	7.981	0.66
10	0.402	0.95	2.974	0.88	4	0.429	0.95	2.574	0.89	20	7.587	0.95	5.453	0.55
20	0.328	0.95	1.428	0.93	8	0.339	0.88	2.005	0.97	30	7.848	0.96	5.129	0.52

## Data Availability

Not applicable.

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
