# Peer review of "Chitosan Fibers Loaded with Limonite as a Catalyst for the Decolorization of Methylene Blue via a Persulfate-Based Advanced Oxidation Process"

_polymers, 2022, doi:10.3390/polym14235165_

Round 1
Reviewer 1 Report (Previous Reviewer 2)
Accept
Author Response
Thank you very much for accepting our manuscript.
Reviewer 2 Report (Previous Reviewer 1)
The manuscript was previously submitted under different title. The authors, as suggested by the reviewers, corrected the Abstract, changed the Conclusions, and corrected a number of editorial errors. However, a few reservations about the content and presentation remained.
The manuscript has a good description of the research material, i.e. chitosan fibers with an admixture of limonite, and a description of the experiments performed. In addition to the data on the kinetics of discoloration, the results of EDX, FTIR, XRD measurements of the tested material are given.
When analyzing the presented results, one gets the impression that this work is part of a larger study.
The paper presents the results for the discoloration of one dye "methylene blue", which is not used in the industry on a mass scale and does not pose a threat to the environment - which was given as the justification for the research in Introduction.
It was not explained why methylene blue was used in the experiments, which is one of the reagents for the pH test. The color change with a change in pH may be confused with the decomposition of the dye under the influence of AOP.
Line 123 - "qm is the maximum adsorption capacity (mg/g)". If Eq. is correct then unit should be "mg/L" - same as unit of qe
Figure 3a. unit of qe should be mg/L, not mg/g
Table 1 - unit for qm should be mg/L, not mg/g. The same is for KF units
Table 2 - "First- and second-order kinetic model parameters" should be "First- and second-order reaction kinetic model parameters"
Figure 4 (b) and (e) - red decolorization curves were intended to show the course of the reaction kinetics for limonite 10% and PS concentratuin 4 mM. However, the red curves are different so one of them is not showing the correct data.
The authors should comment on the shape of the curves in Figure 4. One can see that from 0 to 2 minutes it is a straight line with a different slope than a straight line from 4 to 10 minutes. It is likely that different mechanisms govern these steps in the process.
Figure 4 (c) and (f) - The red lines (limonite 10% and PS concentratuin 4 mM) are identical, but one of the vertical scales is incorrect. The last point is 36 (c) and 30 (f)
Figure 5 (a) - Pleaase, explain why the increase in MB concentration gives the increase in decolorization percentage. Moreover - why after 1 minute the reaction stops. It looks like one of the reactants has run out.
Figure 5 (b) - A sharp jump between pH = 2.7 and pH = 5.78 may be due to a false measurement of the MB concentration by chromatography. It is known that MB is a pH indicator and between these pH values it changes its color which can be interpreted in chromatography as a change in concentration.
Author Response
- The paper presents the results for the discoloration of one dye "methylene blue", which is not used in the industry on a mass scale and does not pose a threat to the environment - which was given as the justification for the research in Introduction. It was not explained why methylene blue was used in the experiments, which is one of the reagents for the pH test. The color change with a change in pH may be confused with the decomposition of the dye under the influence of AOP.
Response:
Methylene blue was chosen as it is one of the most widely used organic dye model compounds in many studies to investigate the performance of wastewater treatment. Statement below was added into the introduction (Line 74).
“Methylene blue was chosen as an organic model compound to investigate the efficiency of the chitosan-limonite fibres.”
We agree with the comments by the reviewer on the concern of pH. In this study, calibration curves of the estimation of MB concentration using UV-vis spectrophotometer were constructed at the same pH used in the experiments. By doing so, we have minimized the errors of the data collected.
- Line 123 - "qm is the maximum adsorption capacity (mg/g)". If Eq. is correct then unit should be "mg/L" - same as unit of qe
Response:
We have double confirmed. The unit of qm is mg/g.
- Figure 3a. unit of qe should be mg/L, not mg/g.
Response:
We have double confirmed. The unit of qe is mg/g, which was calculated using Eq. 1.
- Table 1 - unit for qm should be mg/L, not mg/g. The same is for KF units
Response:
We have double confirmed. The unit of qm is mg/g. Unit for KF is L/g.
We have referred article below:
doi:10.1371/journal.pone.0088802
- Table 2 - "First- and second-order kinetic model parameters" should be "First- and second-order reaction kinetic model parameters"
Response:
Thank you for the comments. Revision was done ad suggested.
- Figure 4 (b) and (e) - red decolorization curves were intended to show the course of the reaction kinetics for limonite 10% and PS concentration 4 mM. However, the red curves are different so one of them is not showing the correct data.
Response:
Thank you for the comment. Figure 4e has been corrected.
- The authors should comment on the shape of the curves in Figure 4. One can see that from 0 to 2 minutes it is a straight line with a different slope than a straight line from 4 to 10 minutes. It is likely that different mechanisms govern these steps in the process.
Response:
We agree with the reviewer. The decolorization of MB could involve different mechanisms, such as adsorption and decomposition.
- Figure 4 (c) and (f) - The red lines (limonite 10% and PS concentration 4 mM) are identical, but one of the vertical scales is incorrect. The last point is 36 (c) and 30 (f)
Response:
Revisions were done as suggested. Thank you for pointing out the mistakes.
- Figure 5 (a) - Please, explain why the increase in MB concentration gives the increase in decolorization percentage. Moreover - why after 1 minute the reaction stops. It looks like one of the reactants has run out.
Response:
Thank you for the comments. We have added further discussion on the effect of MB concentration on the decolorization percentage. The dependence of rate of reaction on concentration of MB and PS was determined to suggest that the decolorization performance is dependence to MB concentration.
- Figure 5 (b) - A sharp jump between pH = 2.7 and pH = 5.78 may be due to a false measurement of the MB concentration by chromatography. It is known that MB is a pH indicator and between these pH values it changes its color which can be interpreted in chromatography as a change in concentration.
Response:
The measurement of the MB concentration was conducted using spectrophotometer, not chromatography. An optical cable probe (FC-UVIR200-2) was immersed into the dye solution without withdrawing aliquots. The probe was connected to a light source (AvaLight-DH-S, Avantes) and a UV-vis spectrometer (AvaSpec-ULS2048L-USB2-UA-RS, Avantes). The obtained data were used to estimate the MB concentration based on a calibration curve constructed from series of MB solutions with different concentrations and pH.

Reviewer 3 Report (New Reviewer)
In this paper, Wongchoosuk et al. report the synthesis of chitosan-limonite fibers loaded with limonite powder and the application of the as-obtained fibers as heterogeneous catalysts for the SO4∙--based AOPs to degrade methylene blue. All the results are clearly presented. The following comments need to be addressed before publication.
Comment 1. (Line 311) In the conclusion, the author claimed “a good distribution of limonite particles throughout the fibers”. However, no data in this manuscript supports this claim. Are those small bright particles in Figure 1 limonite? The EDX elemental mapping of Fe should be provided to help to support this claim.
Comment 2. It is reported that SO4∙- radical can degrade chitosan (Polymer Degradation and Stability, 2002, 75(1): 73-83. https://doi.org/10.1016/S0141-3910(01)00205-1). Did you see any morphology change of the fibers after the decolorization experiment?
Comment 3. (Line 207) Limonite is relatively dense (relatively density: 2.7 to 4.3). Why does a higher content of limonite help to increase MB adsorption? Does limonite have a stronger interaction with MB molecules than chitosan?
Comment 4. (Line 228) The authors attributed the high rate constant of 10% limonite sample to the optimum molar ratio (PS:Fe2O3 = 10:3). If you double the amount of PS added to 20% limonite sample to also make the ratio to 10:3. Would you expect a similar result? Why?
Comment 5. Figure 4d. 8 mM plot and 0 mM could not be distinguished from one another. Consider changing the color of the 8 mM plot.
Comment 6. Limonite is the active component that enables the generation of SO4∙- and the subsequent decolorization of MB. What’s the advantage of the limonite/chitosan composite system compared to limonite? The authors should add another control group, only limonite, for the decolorization and recyclability test. A discussion should be provided to show the necessity of preparing the composite system.
Comment 7. How did you recycle the fibers for the decolorization test? The authors should provide the experimental details.
Author Response
First of all, we would like to thank all reviewers for the valuable comments and suggestions. We have considered each of the comments and made revisions accordingly.
Reviewer 3:
In this paper, Wongchoosuk et al. report the synthesis of chitosan-limonite fibers loaded with limonite powder and the application of the as-obtained fibers as heterogeneous catalysts for the SO4∙--based AOPs to degrade methylene blue. All the results are clearly presented. The following comments need to be addressed before publication.
Comment 1. (Line 311) In the conclusion, the author claimed “a good distribution of limonite particles throughout the fibers”. However, no data in this manuscript supports this claim. Are those small bright particles in Figure 1 limonite? The EDX elemental mapping of Fe should be provided to help to support this claim.
Response:
Thank you for the suggestion. We have conducted EDX analysis on the chitosan-limonite fibers with 10% limonite. The image has been added into Fig. 1.
Comment 2. It is reported that SO4∙- radical can degrade chitosan (Polymer Degradation and Stability, 2002, 75(1): 73-83. https://doi.org/10.1016/S0141-3910(01)00205-1). Did you see any morphology change of the fibers after the decolorization experiment?
Response:
Thanks for the comments. We did not see morphology change of the chitosan-limonite fibers even after 10 cycles of the decolorization experiments. In fact, the fibres were remained intact after the experiments. We believe that the degradation on the chitosan by the SO4-redicals does not affect much on the physical and mechanical properties of the fibres.
Comment 3. (Line 207) Limonite is relatively dense (relatively density: 2.7 to 4.3). Why does a higher content of limonite help to increase MB adsorption? Does limonite have a stronger interaction with MB molecules than chitosan?
Response:
Limonite is known to be in negative charge due to its major iron oxide and hydroxide compounds (goethite and hematite). We have cited a new reference to support this. The sentence (Line 207) has been revised as follow:
“The 20% limonite sample showed the highest maximum adsorption capacity (6.291 mg/g) among all the samples due to the negative charge of limonite, contributing to the adsorption of cationic MB [30].”
Comment 4. (Line 228) The authors attributed the high rate constant of 10% limonite sample to the optimum molar ratio (PS:Fe2O3 = 10:3). If you double the amount of PS added to 20% limonite sample to also make the ratio to 10:3. Would you expect a similar result? Why?
Response:
In this section, the variable parameter studied is the loading amount of limonite powder into the chitosan fibres, i.e., 1, 10, and 20%. We therefore fixed the PS concentration to investigate the decolorization kinetics of MB.
Comment 5. Figure 4d. 8 mM plot and 0 mM could not be distinguished from one another. Consider changing the color of the 8 mM plot.
Response:
Thanks for the suggestion. We have changed the color of the 8 mM plot.
Comment 6. Limonite is the active component that enables the generation of SO4∙- and the subsequent decolorization of MB. What’s the advantage of the limonite/chitosan composite system compared to limonite? The authors should add another control group, only limonite, for the decolorization and recyclability test. A discussion should be provided to show the necessity of preparing the composite system.
Response:
Thank you for the suggestion for us to improve the manuscript. We have conducted additional decolorization experiments using 100% limonite powder with same solid weight of the chitosan-limonite fibers. The results were plotted and supply as supplementary information (Figure S1) as follow:
Figure S1. Decolorization kinetics of MB with 4 mM PS using limonite powder (0.03 g, 0.3 g, and 0.6 g).
It shows similar trend as using the chitosan-limonite fibers, but slightly lower decolorization percentage. This can be attributed to the agglomeration of the limonite particles as they were suspended into the MB solution during the AOP experiment. This observation has further proven that the chitosan helps in avoiding agglomeration of limonite, enhancing the exposed surface of the limonite particles. Besides, immobilization of limonite powder into the chitosan fibres offers other advantages, including avoid leaching of the limonite powder and easy to recollect the fibers especially for the utilization in larger scale, such as packing in fixed bed column.
Sentence below was added into the Introduction (Line 73) to further justify the necessity of preparing the limonite powder into fibre form.
“Immobilization of limonite powder into the chitosan fibres offers several advantages, such as easy recovery and separation of the catalyst [28].”
Additional discussion was added into the manuscript (Line 233):
“Additional decolorization experiments were also conducted using 100% limonite powder (without chitosan) with the same solid weight of the chitosan-limonite fibers, i.e., 0.03 g, 0.3 g, and 0.6 g limonite. The results are shown in Figure S1. It shows similar trend as using the chitosan-limonite fibers, but slightly lower decolorization percentage. This can be attributed to the agglomeration of the limonite particles as they were suspended into the MB solution during the AOPs experiment. This observation has further proven that the chitosan helps in avoiding agglomeration of limonite, increasing the exposed surface area of the limonite particles. Besides, immobilization of limonite powder into the chitosan fibers offers other advantages, including avoid leaching of the limonite powder and easy to recollect the fibers especially for the utilization in larger scale, such as packing in fixed bed column.”
Comment 7. How did you recycle the fibers for the decolorization test? The authors should provide the experimental details.
Response:
Thank you for the suggestion. Sentence below was added into the manuscript at Line 147.
“The reusability of the chitosan-limonite fibres was tested by performing ten consecutive cycles of the SO4-based AOPs experiments using the same experimental set up. The fibres were rinsed three times by deionized water before used for next cycle.”

Round 2
Reviewer 2 Report (Previous Reviewer 1)
In the second version of the manuscript, the authors corrected the errors indicated by the reviewers. However, they did not correct errors not pointed out by the reviewers, which may suggest that the co-authors did not read any version of the submitted manuscript.
In the corrected fragments of the texts, they introduced new linguistic errors.
Line 74-75 - "Methylene blue was chosen as an organic model compound to investigate the efficiency of the chitosan-limonite fibers." The sentence makes no sense. Efficiency of fibers? What does it mean?
Line 76 - "fabrication" change to "application" or "incorporation".
Line 146 - "second-order kinetic models" change to "second-order reaction kinetic models"
Line 149 - Eq (6) - the two terms of the equation on the left are identical, so their difference is zero - which is nonsense. Authors should use larger fonts to see such trivial errors.
Line 149 – “the first and second kinetic models,” change to “the first and second order reaction kinetic models,” – the lack of precision in this term is repeated many times in the paper.
Line 174-175 – “At higher magnifications, the cross-section of the chitosan fiber is more porous” may be changed to “As can be seen at higher magnifications, the cross-section of the chitosan fiber is more porous”
Line 216 - “adsorption capacity (6.291 mg/g)”, however in Figure 3(a) the maximum value is 5.9 ± 0.1
Line 289 – Equation 8 can be omitted as the next one shows the actual results obtained. However, the authors do not explain which data this equation applies to and what correlation coefficient R2 was obtained for these data. Moreover, there is no drawing from which one could read the quality of the fit of Equation 9 to the experimental data.
Line 292 - change "dependence" to "dependent"
Author Response
We would like to sincerely thank the reviewer for the useful comments and suggestions. We have revised the manuscript according to the comments given.
- Line 74-75 - "Methylene blue was chosen as an organic model compound to investigate the efficiency of the chitosan-limonite fibers." The sentence makes no sense. Efficiency of fibers? What does it mean?
Response:
The sentence was revised as follow:
“Methylene blue was chosen as an organic model compound to investigate the decolorizing efficiency of the chitosan-limonite fibers via SO4-×-based AOPs.”
- Line 76 - "fabrication" change to "application" or "incorporation".
Response:
Thank you for the suggestion. The word “fabrication” was changed to “incorporation”.
- Line 146 - "second-order kinetic models" change to "second-order reaction kinetic models".
Response:
Thank you for pointing this mistake. We have changed the “second-order kinetic models” to “second-order reaction kinetic models”.
- Line 149 - Eq (6) - the two terms of the equation on the left are identical, so their difference is zero - which is nonsense. Authors should use larger fonts to see such trivial errors.
Response:
Thank you for pointing out the mistake. The Eq (6) should be:
Besides, we have also used larger font size for the equations.
- Line 149 – “the first and second kinetic models,” change to “the first and second order reaction kinetic models,” – the lack of precision in this term is repeated many times in the paper.
Response:
We have changed all “the first and second kinetic models” to “first- and second-order reaction kinetic models”.
- Line 174-175 – “At higher magnifications, the cross-section of the chitosan fiber is more porous” may be changed to “As can be seen at higher magnifications, the cross-section of the chitosan fiber is more porous”
Response:
Thank you for the suggestion. We have revised the sentence as suggested.
- Line 216 - “adsorption capacity (6.291 mg/g)”, however in Figure 3(a) the maximum value is 5.9 ± 0.1.
Response:
The value of adsorption capacity (qm) was obtained from the fitting of adsorption data into the linearized Langmuir adsorption isotherm equation (Eq. 2). It is calculated from the slope of the linear line in Fig. 3b. Therefore, it should have slight deviation with the non-linearized plot.
- Line 289 – Equation 8 can be omitted as the next one shows the actual results obtained. However, the authors do not explain which data this equation applies to and what correlation coefficient R2was obtained for these data. Moreover, there is no drawing from which one could read the quality of the fit of Equation 9 to the experimental data.
Response:
Equation 8 was removed as suggested.
The determination of reaction order is based on the obtained kinetic rate constants shown in Table 2. Hence, there are no correlation coefficients obtained because the calculations were based on substitution method by using equation below:
Rate = k[MB]x[PS]y
In addition, we have also added the fitted plots of the first- and second-order reaction kinetic models for the MB decolorization at different MB concentrations (Figure 5 (b&c)). The obtained rate constants (k1 & k2) and R2 were also added into Table 2.
- Line 292 - change "dependence" to "dependent"
Response:
Revision was done as suggested. Thank you.

Reviewer 3 Report (New Reviewer)
Thank you for addressing my comments. I recommend this manuscript be accepted in its present form.
Author Response
Thank you for the recommendation. We appreciate your input and comments given.

Round 3
Reviewer 2 Report (Previous Reviewer 1)
Two notes on authors' responses:
7. Line 216 - “adsorption capacity (6.291 mg/g)”, however in Figure 3(a) the maximum value is 5.9 ± 0.1.
Response:
The value of adsorption capacity (qm) was obtained from the fitting of adsorption data into the linearized Langmuir adsorption isotherm equation (Eq. 2). It is calculated from the slope of the linear line in Fig. 3b. Therefore, it should have slight deviation with the non-linearized plot.
My comment: 5.9 is the measured value – it is real. 6.291 is not real value – it exists on paper – it is virtual. IMHO put here the really measured value
“The determination of reaction order is based on the obtained kinetic rate constants shown in Table 2. Hence, there are no correlation coefficients obtained because the calculations were based on substitution method by using equation below:
Rate = k[MB]x[PS]y "
My comment: I don't know how you used the "substitution method" and which exact data was used for the calculations. In chemical engineering, for z = a*xb*yc form of equations, regression analysis is used for the transformed equation log(z) = log(a) + b*log(x) + c*log(y).
In many papers, researchers use the ANOVA package. It also can be done in Excel.
Line 304-305 - "The experiments is more fitted with first order reaction kinetic model and the reaction rate increased with increasing MB concentration (as shown in Table 2)." - change to "is better fitted"
It is very good that the authors have added Figures 5b and 5c. Of course, in Figure 5a, the time scale can be successfully limited to 2 minutes. In Figure 5, the inscription "Decolorization percentage" should rather be "Decolorization efficiency" and similarly in the text to maintain uniform nomenclature.
P.S.
Note to Figures 5b and 5c. The term ln(Ct-Ce)/(C0-Ce) at time zero is 1, so ln(1) = 0. Meanwhile, in the graph, the data starts in the range 1 to 2.5.In Figure 5c 1/(Ct-Ce) - 1/(C0-Ce) for time zero, both terms are equal and their difference is 0. Meanwhile, the points on the graph have initial values of about minus 0.5
There is no reference to any article in the description in point 3.3. If the information in this section is important for the interpretation of the results, the authors should write something about it. Without such explanations, it seems that Figure 6 was included just because it's pretty.
No point 4. Point 3.2 is followed by figure 6 and point '5. Conclusions
Author Response
Many thanks for the comments and suggestion given. We have revised the manuscript accordingly.
Two notes on authors' responses:
- My comment: 5.9 is the measured value – it is real. 6.291 is not real value – it exists on paper – it is virtual. IMHO put here the really measured value.
Response:
We have changed the adsorption capacity (qm) to 5.9 mg/g as suggested (Line 217).
- “The determination of reaction order is based on the obtained kinetic rate constants shown in Table 2. Hence, there are no correlation coefficients obtained because the calculations were based on substitution method by using equation below:
Rate = k[MB]x[PS]y "
My comment: I don't know how you used the "substitution method" and which exact data was used for the calculations. In chemical engineering, for z = a*xb*yc form of equations, regression analysis is used for the transformed equation log(z) = log(a) + b*log(x) + c*log(y).
In many papers, researchers use the ANOVA package. It also can be done in Excel.
Response:
Thank you for the suggestion. We have used the transformed equation: log Rate = log k + a log [MB] + b log [PS] as suggested. For the fitting of [MB], [PS] was set as constant, and vice versa. The plots are shown as follows:
The reaction order for both [MB] and [PS] were obtained from each plots, and are in positive value, suggested that the reaction of SO4-×-based AOPs is concentration dependent for both MB and PS.
- Line 304-305 - "The experiments is more fittedwith first order reaction kinetic model and the reaction rate increased with increasing MB concentration (as shown in Table 2)." - change to "is better fitted"
It is very good that the authors have added Figures 5b and 5c. Of course, in Figure 5a, the time scale can be successfully limited to 2 minutes. In Figure 5, the inscription "Decolorization percentage" should rather be "Decolorization efficiency" and similarly in the text to maintain uniform nomenclature.
Response:
Thank you for the suggestion. We have revised the sentence “… is more fitted..” to “… is better fitted..”.
The time scale of Fig. 5(a) has been changed and limited to 2 min. In addition, the inscription “Decolorization percentage” has been changed to “Decolorization efficiency”.
- Note to Figures 5b and 5c. The term ln(Ct-Ce)/(C0-Ce) at time zero is 1, so ln(1) = 0. Meanwhile, in the graph, the data starts in the range 1 to 2.5.
In Figure 5c 1/(Ct-Ce) - 1/(C0-Ce) for time zero, both terms are equal and their difference is 0. Meanwhile, the points on the graph have initial values of about minus 0.5
Response:
Many thanks for the comment. We have double checked the fitting and revised the Figure 5b and 5c.
- There is no reference to any article in the description in point 3.3. If the information in this section is important for the interpretation of the results, the authors should write something about it. Without such explanations, it seems that Figure 6 was included just because it's pretty.
Response:
A new reference (ACS Appl. Mater. Interfaces 2020, 12, 50) was cited in point 3.3 to further support the discussion on the adsorption energy of PS to the surface of the limonite (hematite and goethite). The more negative Eads value the stronger the adsorption is.
Figure 6 is important to illustrate the configuration of the crystal slab of hematite and goethite, as well as to show how the arrangement of PS molecules and the crystal plane of hematite and goethite. Therefore we would like to remain it in the manuscript.
- No point 4. Point 3.2 is followed by figure 6 and point '5. Conclusions
Response:
Point 5 for the conclusion has been changed to Point 4.

This manuscript is a resubmission of an earlier submission. The following is a list of the peer review reports and author responses from that submission.
Round 1
Reviewer 1 Report
Review
The manuscript has a good description of the research material, i.e. chitosan fibers with an admixture of limonite, and a description of the experiments performed. In addition to the data on the kinetics of discoloration, the results of EDX, FTIR, XRD measurements of the tested material are given. The conclusions presented in the summary are supported by the results of the experiments.
When analyzing the presented results, one gets the impression that this work is part of a larger study.
1) The title says "organic dyes", however, the paper presents the results only for the discoloration of one dye "methylene blue", which is not used in the industry on a mass scale and does not pose a threat to the environment - which was given as the justification for the research in Introduction (line 38-39). I would suggest clarifying the title of the paper.
2) It was not explained why methylene blue was used in the experiments. It is one of the reagents for the pH tests. The color change with a change in pH may be confused with the decomposition of the dye under the influence of AOP.
Detailed editorial notes:
Line 31 - change "under" to "after"
Line 72 - after "methylene blue" add "(MB)"
Line 96 - do not use abbreviations in subsection titles - change "MB" to "methylene blue"
Line 106 - add units to Ce, qe
Line 130, 134 and next - "first and second kinetic models" replace with "first- and second-order kinetic models", or "first- and second-order reaction models", as the term "order" is attributed to reaction, and not kinetics.
Line 155-157 - Sentence should be reformulated to increase clarity - may add "one can see that the cross-section"
Figure 4 (b) and (e) - red decolorization curves were intended to show the course of the reaction kinetics for limonite 10% and PS concentratuin 4 mM. However, the red curves are different so one of them is not showing the correct data.
Figure 4 (c) and (f) - The red lines (limonite 10% and PS concentratuin 4 mM) are identical, but one of the vertical scales is incorrect. The last point is 36 on (c) and 30 on (f)
Figure 5 (a) - explain why the increase in MB concentration gives the increase in decolorization percentage. Moreover - why after 1 minute the reaction stops. It looks like one of the reactants has run out.
Figure 5 (b) - A abrupt jump between pH = 2.7 and pH = 5.78 may be due to a false measurement of the MB concentration by chromatography. It is known that MB is a pH indicator and somewhere between these pH values it changes its color which can be interpreted as a change in concentration.
Figure 5 (c) - the caption under the figure does not agree with the numbering on the horizontal scale - the first bar should be numbered 2, or the first bar of 100% should be added.
For readability, the abbreviations MB and PS should be written in full in Conclusions
And the last question: were the pH and temperature monitored during each experiment ?. Nothing is mentioned in the work on this subject.
Reviewer 2 Report
Detailed comments:
1. The English of the text should be checked
2. The novelty of manuscript should be highlighted more
3. Please eliminate multiple references. After that, please check the manuscript thoroughly and eliminate ALL the lumps in the manuscript. This should be done by characterizing each reference individually and by mentioning 1 or 2 phrases per reference to show how it is different from the others and why it deserves mentioning. Multiple references are of no use for a reader and can substitute even a kind of plagiarism, as sometimes authors are using them without proper studies of all references used. In the case, each reference should be justified by it is used and at least short assessment provided.
4. In the Introduction part, the authors must be included new, relevant and more information about AOPs method, limonite (advantages and disadvantages, in comparison with other materials that contains hydrated iron oxides). Also, information about chitosan must be included (advantages, disadvantages, applications). The following references can be included in the Introduction part to improve the quality of manuscript, because they provide relevant information:
ü Fe3O4@HAP-enhanced photocatalytic degradation of Acid Red73 in aqueous suspension: Optimization, kinetic, and mechanism studies. Mater. Res. Bull. 2017, 91, 59–67
ü Modified Composite Based on Magnetite and Polyvinyl Alcohol: Synthesis, Characterization, and Degradation Studies of the Methyl Orange Dye from Synthetic Wastewater. Polymers 2021, 13(22), 3911
ü Photocatalytic pathway on the degradation of methylene blue from aqueous solutions using magnetite nanoparticles. J. Clean. Prod. 2021, 318, 128556
ü Investigation of an adsorbent based on novel starch/chitosan nanocomposite in extraction of indigo carmine dye from aqueous solutions. Biointerface Res. Appl. Chem. 2020, 10, 5556–5563
ü Biopolymeric membrane enriched with chitosan and silver for metallic ions removal, Polymers, 2020, 12(8), 1792
5. At 2.2 Preparation of chitosan fibers loaded with limonite - a schematically representation of preparation of the materials must be included
6. All equipment and tools used in this study should be described in detail or further information should be provided (manufacturer, type, operational conditions, etc).
7. At lines 81-82, authors write: “…then stirred for 1 hour to produce a homogeneous suspension.” – indicate in which device the stirring was carried out, the stirring speed
8. At lines 83-84, authors write: “…then the solution was injected continuously into 0.1 M NaOH solution…” – what means injected continuously (the time or period must be indicated) and, the amount for NaOH solution must be indicated
9. For all parameters indicated in all equations, unit of measure must be indicated
10. At Figure 3b, for Ce, at scale, unit of measure must be indicated
11. At Figure 5a, the point must be indicated; the indicated curves are not enough (the points between which the related curves were drawn must also be indicated). Also, at Figures 5b and 5c, for each column must be indicated the values with the value for standard deviation
12. More Conclusions must be included
13. Comparison between the obtained results and measured in this study with other reported studies should be done and included for more clarity (indicate values not just number of reference).
14. Other possible applications of the prepared material must be included at the Conclusion part
15. Same Reference are very old. The manuscript must contain the relevant information to be attractive for readers (researchers), because science has advanced, and the information indicated in the manuscript is no longer valid. This part should include observed information, noted in the last 10-12 years.
Reviewer 3 Report
In this work, the limonite-loaded chitosan fibers were fabricated as catalysts. And their morphology and structure, adsorption and decolorization abilities of methylene blue, stability and reusability, and computational adsorption energy calculation were discussed in detail. However, there are still some issues to be addressed:
1. There are many grammar and format mistakes. Please check the whole manuscript carefully.
2. In the Abstract, Lines 21, the full name of “COD” should be indicated.
3. In the Introduction, the innovation of your work should be emphasized.
4. It is suggested to add a process flow diagram on the material preparation.
5. Lines 96, please explain the differences between adsorption and decolorization.
6. It is suggested to compare the properties of your material with those of other literatures.
7. The analysis of adsorption thermodynamics should be provided.
Reviewer 4 Report
Lines 83-86.” …then the solution was injected continuously into 0.1 M NaOH solution to yield regenerated chitosan-limonite fibers. Next, the resulting fibers were washed with deionized water three times to remove excess chemicals. ” At what rate was the solution squeezed out of the syringe? What was the fiber volume/flush water ratio during fiber cleaning?
Lines 196-198. :Chitosan fibers with limonite content greater than 20% were not successfully produced due to the difficulty of the injection using the syringe needle. “ Given that the resulting "fiber" was clearly not of substantial length and not suitable for making bundles of fibers or fabrics, why don't the authors try producing a "scaled" composite, by depositing it in the same sodium hydroxide or by depositing it with organic solvents?
Lines 208-209, 250, 255, 290. “The decolorization percentage of MB for the chitosan fibers with 1, 10, and 20% limonite content were of 73.1, 78.8, and 75.0%, respectively.” I suggest that the authors round up the values to integers. Fractional percent accuracy, strictly speaking, requires multiple repetitions of the experiment and accurate measurements when conducting these repetitive experiments. The same remark applies to the data given in the abstract.